# Predictors of diarrhea among children under 24 months in Nepal: A multilevel analysis of multiple indicator cluster survey 2019

Bipin Thapa[1]*, Suman Sapkota[2], Yifei Hu[2]

**1** Department of Research and Development, Dhulikhel Hospital-Kathmandu University Hospital, Kavre, Nepal, **2** Department of Child and Adolescent Health and Maternal Care, School of Public Health, Capital Medical University, Beijing, China

* bipinthapa2050@gmail.com

## Abstract

### Background

Diarrhea has remained an unmet challenge in global child health. Its severity may be worse than reported in resource-limited settings. Understanding changing epidemiology is critical to combat diarrheal morbidity. Therefore, this study aimed to understand factors associated with diarrhea among under two years children in Nepal.

### Methods

A total of 2,348 samples from Multiple Indicator Cluster Survey 2019 were used to estimate the significant child, maternal, household and external environmental predictors of diarrhea using multilevel analysis.

### Results

The prevalence of diarrhea was 11.9% (95% CI: 10.2%-13.6%). Children residing in Koshi Province [AOR (Adjusted Odds Ratio): 2.23, 95% CI: 1.22–4.08], Karnali Province (AOR: 2.28, 95% CI: 1.11–4.70), and Sudurpaschim Province (AOR: 4.49, 95% CI: 2.39–8.42) were at higher risk of diarrhea. Children aged 7–23 months (AOR: 1.56, 95% CI: 1.10–2.20), children with ARI symptoms (AOR: 4.14, 95% CI: 2.21–7.72) and children whose mothers had no access to prenatal care (AOR: 1.87, 95% CI: 1.01–3.45) had a higher risk of diarrhea. Children from below the richest household wealth group (AOR: 1.76, 95% CI: 1.01–3.08) and those from households practicing open defecation, with unimproved or limited sanitation facilities (AOR: 1.52, 95% CI: 1.09–2.11) were more likely to have diarrhea.

### Conclusion

The findings underscore the need for public health policy-makers to improve sanitation facilities, especially focusing on poor households from Karnali and Sudurpaschim Provinces practicing open defecation to protect the children from the life risk of diarrhea in Nepal.

**Data Availability Statement:** Data are available upon reasonable request. Data are available in a public, open access repository. We used publicly available data and it is accessible from the MICS

website (https://mics.unicef.org/surveys) on the
request.

**Funding:** The author(s) received no specific
funding for this work.

**Competing interests:** The authors have declared
that no competing interests exist.

## Background

Diarrhea is a dreadful matter of concern, accounting for one in ten fatalities (9.1%) in children under the age of five years in 2019 [1], particularly in children under the age of 24 months [2]. Diarrheal morbidity is concentrated much more in underprivileged communities, within resource-limited countries [3, 4]; almost 90% (89.37%) of diarrheal deaths occurred in South Asia and Sub-Saharan Africa [5]. Although total global deaths from diarrhea have decreased over the last several decades, this reduction has not shown a similar decline across locations [3, 5]. Moreover, concerns are emerging about the comeback of diarrhea-associated mortality [6, 7]. Diarrhea has remained an unmet challenge in global child health posing a multitude of consequences, including stunted growth, malnutrition, increased vulnerability to other infectious diseases, and even death [8, 9].

Nepal has a high diarrheal morbidity rate among children under the age of five [10, 11]. This proportion of morbidity is even higher among children under the age of two. According to the 2016 Nepal Demographic and Health Survey (NDHS), the prevalence of diarrhea is 10% among children under two years of age [12]. As a resource-limited country, 18.5% of Nepalese people live below the poverty line [13] and over half of the population (53.2%) don't have access to piped drinking water [14]. There are no safe toilets in 9.8% of the urban areas and 25.0% of the rural areas. Especially, in 43.5% of the poorest household quintiles, there are no toilet facilities [14]. Proper disposal of children's feces is critical to prevent the spread of infections. Nevertheless, more than half (54%) of under two children's mothers had their babies' stools unsafely disposed of [12]. Therefore, diarrheal prevalence and severity may be worse than that reported in resource-limited settings like Nepal.

The environment where a child live is primarily controlled by and experienced via the mother, therefore child diarrhea needs to be understood through the mother. A mother's ability to care for her child is influenced by the family's contextual factors. Global studies have shown child factors such as age, immunization and nutritional status, and presence of comorbidities have an association with the presence of diarrheal morbidity [15–18]. Likewise, maternal factors such as education, media exposure, prenatal examination [19–21] and family contextual factors such as income, water, sanitation and hygiene (WASH) behaviors and spatial factors [19, 22] showed a relation with child diarrhea. Previous studies from Nepal have related diarrheal morbidity with child's nutritional factors, maternal educational factors, and household's economic, sociocultural, and WASH-related behaviors [23–27].

Identification of variables in child and mother's life and their contextual environments is crucial to prevent child diarrhea. Continuously monitoring and understanding changing diarrheal epidemiology is critical for developing effective public health interventions to lower diarrheal incidence, consequences, and pressure on the health system. Limited studies have explored the diarrheal risk factor among under five children in Nepal. Majority of these studies were constrained to specific geographic areas and ignored the effect of geographic heterogeneity [25–27] or utilized NDHS data up to 2016 [23]. To the extent of our knowledge, we did not find national-level studies that explicitly explored diarrheal risk factors among under two years children in Nepal. Therefore, this study aimed to understand the factors associated with diarrhea among under two years children in Nepal using nationally representative data from the latest Multiple Indicator Cluster Survey (MICS) 2019. The findings can be used to inform public health policy-makers and practitioners to prevent and protect children from diarrheal infections.

## Materials and methods

### Data sources

This study was based on the data from MICS 2019, which was conducted by the Central Bureau of Statistics (CBS) in collaboration with the United Nations Children's Fund (UNICEF). The Nepal MICS 2019 was designed to provide estimates for many indicators on the situation of children and women at the national level, for urban and rural areas of seven provinces. A multistage (2-stage) sampling technique was used. The urban and rural areas within each province were made the main sampling strata. Rural municipalities represent the rural area whereas the municipalities, sub-metropolitan, and metropolitan cities represent the urban area. The urban part of Kathmandu valley was included as a separate stratum. Within each stratum, primary sampling units (PSUs) were selected systematically with probability proportional to size. A household listing was conducted within the selected PSUs, identifying the households with and without children under 5 years. In total, 25 households with and without children under 5 were selected in each PSU through a systematic random sampling method. To ensure strong representativeness of children under 5 in the sample, households with children under 5 were oversampled, where 13 households with children under 5 and 12 households without were selected from the listing in each Enumeration Area (EA) [28]. The website of the MICS program makes the anonymized survey dataset freely available under request [29].

### Participants and sample size

The overall sample size was 12,800 households with 512 EAs. The details of the calculation have been reported in the MICS Nepal 2019 report under Appendix A [28]. A total of 14,805 women of age 15–49 years were interviewed, which included 6,658 mothers/caretakers of children under 5 years. After downloading the dataset, observations of children above 23 months were excluded and 2,566 observations out of 6,658 mothers/caretakers were retained. Furthermore, 218 observations for different variables with missing data were removed and 2,348 observations were included in the final analysis.

### Conceptual framework

We categorized the determinants of diarrhea into external environmental factors and child, maternal and household factors based on an extensive literature review and variables available in the dataset. The conceptual framework for the study is shown in Fig 1.

### Study variables

The occurrence of diarrhea in the last two weeks was the dependent variable, whereas child, maternal, household and external environmental characteristics of the children were included as the independent variables. The description and measurement of the study variables are presented in Table 1.

### Statistical methods

**Data management.** After approval of the request by UNICEF, data were received via e-mail in Statistical Package for Social Science (SPSS version 23) format. Three datasets namely household dataset, child dataset and women dataset were used for this study. Required variables from the household and women datasets were merged into the child dataset. Variables were then created and categorized as per the objectives of the study using SAS OnDemand for Academics and syntax and output files were documented.

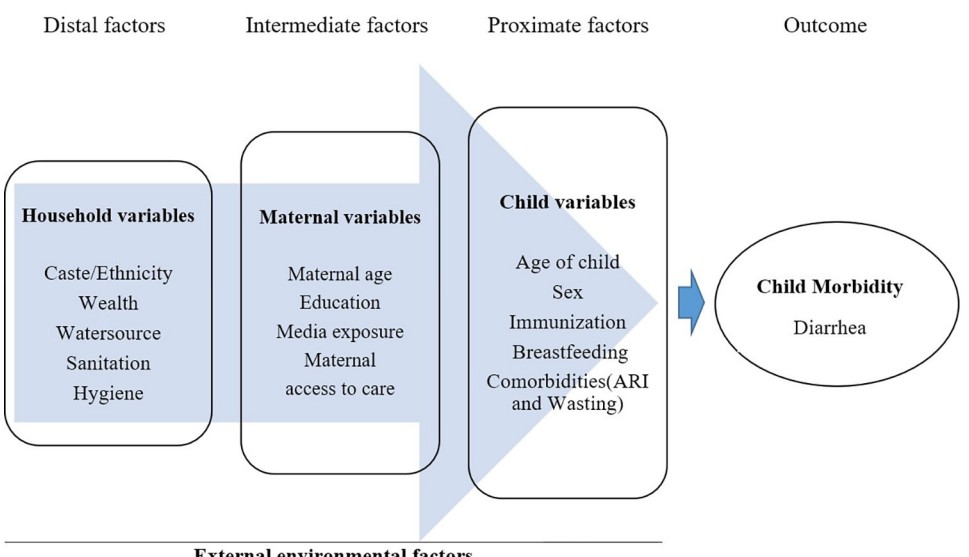

**Fig 1. Conceptual framework for diarrheal risk among children under 24 months.**

**Data analysis.** Descriptive statistics were reported for all variables. Weight, cluster and strata were adjusted to calculate the descriptive statistics. First, bivariate analysis was conducted where the Rao-Scott chi-square test was applied to determine the p-value between diarrhea and independent variables, and variables having p-value less than 0.2 in bivariate analysis were considered candidate variables for the multilevel analysis [31]. The multilevel analysis comprised 2,348 individuals (level 1) nested into 512 PSUs (level 2). Multilevel analyses were performed where child, maternal and household factors were considered in level 1 whereas external environmental factors were placed in level 2. The estimation of generalized linear mixed models (GLMM), including random effects and model fitness, was carried out using the GLIMMIX procedure adjusting for the effects of sampling weight and clustering. Three models were built: model 0 for the null model with PSU as a random parameter and no fixed parameters, model 1 with external environmental variables including the null model, and final model (model 2) with child, maternal and household-level variables including model 1 variables. Covariance parameters were significant for all models. The multicollinearity of the variables was tested before entering into models. No multicollinearity was seen among the independent variables. Adjusted Odds Ratios (AOR) with 95% CI were calculated to determine the significant predictors of diarrhea.

## Ethics

The survey protocol was approved by the CBS as per the Statistical Act (1958) in September 2018. The Statistical Act enables CBS to conduct surveys according to the government's ethics protocol without involving an institutional review board.

## Results

The prevalence of diarrhea among children was 11.9% (95% CI: 10.2%-13.6%). The median (third quartile, first quartile) age of children was 11.0 (18.0, 6.0) months. Almost three-fourths (73.8%) of the children were 7–23 months old. The number of boys (53.8%) was slightly higher

**Table 1. Study variables.**

| Variables | |
|---|---|
| **Dependent variable** | |
| **Diarrhea in the last two weeks** | Yes: Mothers or caretakers reported that their child had an episode of diarrhea in the two weeks prior to the survey. |
| **Independent variables** | |
| **Child characteristics** | |
| **Age of child** | 0–6 months, 7-24months. |
| **Sex of child** | Male and female. |
| **Ever vaccinated** | No: Child was not vaccinated or didn't have any vaccination documents when asked. |
| **Current breastfeeding** | Still breastfed: Child who had ever breastfed and continuing breastfeeding. |
| | Discontinued: Child who had ever breastfed but had discontinued at the time of the survey. |
| | Never breastfed: Child who was reported having no breastfeeding experience. |
| **ARI** | ARI Present: Child who had an illness with a cough with rapid or difficult breathing, and whose symptoms were perceived to be due to a problem in the chest or both a problem in the chest and a blocked or runny nose. |
| **Wasting** | Wasting Present: Child whose Z-score for Body Mass Index (BMI) is < -2 standard deviations below the median. |
| **Maternal characteristics** | |
| **Age of mother** | Complete age of mothers in years divided into three categories:15–24 years, 25–34 years and 35 years and above. |
| **Education status of mother** | Educated: Mother ever attended formal education. |
| **Media exposure** | No: Mother not exposed to the television and radio and newspaper. |
| **Received prenatal care** | Yes: Mother who had received prenatal care prior to child delivery. |
| **Household characteristics** | |
| **Caste ethnic group** | Relatively advantaged: Brahmin, Chhetri, Newar, Magar, Gurung and Thakuri. |
| | Relatively less advantaged: Other castes [30]. |
| **Wealth group** | The survey households according to the wealth divided into five equal parts as Poorest, Second, Middle, Fourth and Richest. For this research, Poorest to fourth quintiles are referred as the below richest group and the fifth quintile is referred to as richest group. |
| **Water source** | Improved water: Drinking water from an improved source. |
| | Unimproved water: Drinking water from an unprotected dug well or unprotected spring. |
| | Surface water: Drinking water directly from a river, dam, lake, pond, stream, canal or irrigation canal. |
| **Sanitation** | Basic: Use of improved facilities that are not shared with other households. |
| | Limited: Use of improved facilities shared between two or more households. |
| | Unimproved: Use of pit latrines without a slab or platform, hanging latrines or bucket latrines. |
| | Open defecation: Disposal of human faces in fields, forests, bushes, open bodies of water, beaches or other open spaces, or with solid waste. |
| **Hygiene** | Basic: Availability of a hand washing facility on premises with soap and water. |
| | Limited: Availability of hand washing facility on premises without soap and water. |
| | No facility: No hand washing facility on premises. |
| **External environmental characteristics** | |
| **Area** | Urban: Urban municipality, Sub-metropolitan city and Metropolitan city. |
| | Rural: Rural municipality. |
| **Province** | Koshi Province, Madhesh Province, Bagmati Province, Gandaki Province, Lumbini Province, Karnali Province, Sudurpaschim Province. |

than the number of girls. The proportion of immunized children was 89.1%. Around 94% of the children were still breastfeeding while 2.8% of children discontinued breastfeeding and the same percentage of children were never breastfed. Nearly 3% of the children showed the symptoms of ARI and over one-tenth of children (13.6%) were wasted. The median (third quartile, first quartile) age of mothers/caretakers was 25.0 (29.0, 22.0) years. Mothers/caretakers aged 15–24 and 25–34 years old accounted for nearly 46% each, and 78.8% of mothers/caretakers were educated. Similarly, most of the mothers had media exposure (71.7%) and had received prenatal care (95.8%). Around, three in five (61.5%) children were from a relatively less advantaged ethnic caste group. Similarly, slightly more than 80% of the children were from the below-richest wealth group. Regarding children's WASH components, almost 98% of the household had been drinking water from an improved source. More than three-fourths of the households had access to basic sanitation (76.5%). Likewise, basic hygiene was practiced by 79.0% of the households. Nearly about 65% of the households were from urban areas. Most of the households were from the Madhesh Province (21.7%) (Table 2).

The association of various factors with diarrhea is shown in Table 3. Maternal and child factors with p-value less than 0.2 were age of child, sex of child, presence of ARI symptoms and received prenatal care. Likewise, household factors like the wealth group, sanitation and hygiene had a p-value less than 0.2. Similarly, external environmental factors like area and Province had p-value less than 0.2 and these variables were considered candidate variables for multilevel analysis (Table 3).

Multilevel models of diarrhea are presented in Table 4. The effect of external environmental factors on diarrhea prevalence is depicted in model 1, where Koshi Province, Karnali Province, and Sudurpaschim Province had a significant association with diarrhea.

In the final model, from an external environmental perspective, those children residing in Koshi Province (AOR: 2.23, 95% CI: 1.22–4.08), Karnali Province (AOR: 2.28, 95% CI: 1.11–4.70), and Sudurpaschim Province (AOR: 4.49, 95% CI: 2.39–8.42) were at higher risk of diarrhea. Maternal and child factors like children aged 7–23 months (AOR: 1.56, 95% CI: 1.10–2.20), presence of ARI symptoms (AOR: 4.14, 95% CI: 2.21–7.72) and no access to prenatal care (AOR: 1.87, 95% CI: 1.01–3.45) were associated with diarrhea. Children from household's wealth group below the richest (AOR: 1.76, 95% CI: 1.01–3.08), practicing open defecation, with unimproved or limited sanitation facilities (AOR: 1.52, 95% CI: 1.09–2.11) were more likely to have diarrhea (Table 4).

## Discussion

This study was undertaken to understand the child, maternal, household and external environmental factors associated with diarrhea among under two years children in Nepal. We found that the prevalence of diarrhea was 11.9% (95% CI: 10.2%-13.6%). Further, children of age 7–23 months, children with ARI symptoms, children with mothers without prenatal care, children from poorer households, children from households practicing open defecation or limited or unimproved sanitation facility and children residing in Koshi Province, Karnali Province and Sudurpaschim Province were more vulnerable to diarrhea infection.

### Association with child, maternal and household-level variables

We found that the older children aged 7–23 months compared to 0–6 months of age were more likely to have diarrhea. This finding is consistent with findings from previous study in Nepal and other several global studies [22, 24, 32, 33]. Children of the ages of 0–6 months in Nepal are usually exclusively breastfed [28], and it has been proven that exclusive breastfeeding for the first six months of a child's life plays a conclusive protective role in preventing rotavirus

**Table 2. Characteristics of children under 24 months.**

| Characteristics | | n(%) |
|---|---|---|
| **Diarrhea in the last two weeks** | Yes | 280(11.9) |
| | No | 2068(88.1) |
| **Age of child in months** | 0–6 | 615(26.2) |
| | 7–23 | 1733(73.8) |
| **Sex of child** | Male | 1263(53.8) |
| | Female | 1085(46.2) |
| **Ever vaccinated** | Yes | 2093(89.1) |
| | No | 255(10.9) |
| **Current breastfeeding** | Still breastfeeding | 2217(94.4) |
| | Discontinued | 65(2.8) |
| | Never breastfed | 66(2.8) |
| **ARI** | Present | 66(2.8) |
| | Absent | 2282(97.2) |
| **Wasting** | Present | 320(13.6) |
| | Absent | 2028(86.4) |
| **Age of mother in years** | 15–24 | 1091(46.5) |
| | 25–34 | 1087(46.3) |
| | 35 and above | 170(7.2) |
| **Education status of mother** | Educated | 1849(78.8) |
| | None | 499(21.2) |
| **Media exposure** | Yes | 1684(71.7) |
| | No | 664(28.3) |
| **Received prenatal care** | Yes | 2249(95.8) |
| | No | 99(4.2) |
| **Caste ethnic group** | Relatively advantaged | 904(38.5) |
| | Relatively less advantaged | 1444(61.5) |
| **Wealth group** | Below Richest | 1947(82.9) |
| | Richest | 401(17.1) |
| **Water source** | Improved | 2306(98.2) |
| | Unimproved | 39(1.7) |
| | Surface water | 3(0.1) |
| **Sanitation** | Basic | 1795(76.5) |
| | Limited | 425(18.1) |
| | Unimproved | 8(0.3) |
| | Open defecation | 120(5.1) |
| **Hygiene** | Basic | 1855(79.0) |
| | Limited | 454(19.4) |
| | No facility | 39(1.6) |
| **Area** | Urban | 1522(64.8) |
| | Rural | 826(35.2) |
| **Province** | Koshi | 366(15.6) |
| | Madhesh | 510(21.7) |
| | Bagmati | 454(19.3) |
| | Gandaki | 178(7.6) |
| | Lumbini | 459(19.5) |
| | Karnali | 154(6.6) |
| | Sudurpaschim | 227(9.7) |

**Table 3. Association of independent variables with diarrhea among children under 24 months.**

| Factors | | Diarrhea n(%) | No Diarrhea n(%) | p-value |
|---|---|---|---|---|
| **Age of child** | 7–23 months | 225(13.0) | 1508(87.0) | 0.03 |
| | 0–6 months | 55(8.9) | 560(91.1) | |
| **Sex of child** | Male | 162(12.8) | 1101(87.2) | 0.19 |
| | Female | 118(10.9) | 967(89.1) | |
| **Ever vaccinated** | No | 31(12.0) | 224(88.0) | 0.97 |
| | Yes | 250(11.9) | 1843(88.1) | |
| **Current breastfeeding** | Never breastfed/ Discontinued | 14(10.7) | 117(89.3) | 0.66 |
| | Still breastfed | 266(12.0) | 1951(88.0) | |
| **ARI** | Present | 24(35.8) | 42(64.2) | <0.001 |
| | Absent | 257(11.2) | 2025(88.8) | |
| **Wasting** | Present | 35(10.9) | 285(89.1) | 0.64 |
| | Absent | 245(12.1) | 1783(87.9) | |
| **Age of mother** | 15–24 years | 137(12.5) | 955(87.5) | 0.70 |
| | 25–34 years | 126(11.6) | 961(88.4) | |
| | 35 years and above | 17(10.1) | 152(89.9) | |
| **Education status of mother** | None | 56(11.3) | 443(88.7) | 0.67 |
| | Educated | 224(12.1) | 1625(87.9) | |
| **Media exposure** | No | 85(12.8) | 579(87.2) | 0.51 |
| | Yes | 195(11.6) | 1489(88.4) | |
| **Received prenatal care** | No | 18(18.6) | 81(81.4) | 0.11 |
| | Yes | 262(11.6) | 1987(88.4) | |
| **Caste ethnic group** | Relatively less advantaged | 170(11.8) | 1274(88.2) | 0.80 |
| | Relatively advantaged | 110(12.2) | 794(87.8) | |
| **Wealth group** | Below richest | 258(13.3) | 1689(86.7) | <0.001 |
| | Richest | 22(5.5) | 379(94.5) | |
| **Water source** | Unimproved/surface water | 7(17.2) | 35(82.8) | 0.25 |
| | Improved | 273(11.8) | 2033(88.2) | |
| **Sanitation** | Limited/ Unimproved/open defecation | 77(13.9) | 476(86.1) | 0.19 |
| | Basic | 204(11.3) | 1591(88.7) | |
| **Hygiene** | Limited/ No facility | 68(13.9) | 424(86.1) | 0.17 |
| | Basic | 212(11.4) | 1644(88.6) | |
| **Area** | Rural | 120(14.5) | 706(85.5) | 0.02 |
| | Urban | 160(10.5) | 1362(89.5) | |
| **Province** | Koshi | 59(16.2) | 306(83.8) | <0.001 |
| | Madhesh | 44(8.7) | 466(91.3) | |
| | Gandaki | 20(11.0) | 159(89.0) | |
| | Lumbini | 47(10.2) | 413(89.8) | |
| | Karnali | 23(15.0) | 131(85.0) | |
| | Sudurpaschim | 58(25.7) | 168(74.3) | |
| | Bagmati | 29(6.5) | 425(93.5) | |

diarrhea [34]. Another reason could be that children above the age of six months begin crawling or walking, increasing their exposure to an infectious pathogen. Such agents can be transferred via touch [35]. Moreover, such children start complementary feeding after the age of 6 months, which may increase the risk of diarrhea [24, 36] as they could be exposed to diarrheal agents via contaminated food and water [35]. Evidence suggests that the incidence of diarrhea among children appears highest during the weaning period when complementary foods are administered [35, 37].

**Table 4. Predictors of diarrhea among children under 24 months using multilevel models.**

| Factors | | Model 1 | Model 2[†] |
|---|---|---|---|
| | | AOR (95% CI) | AOR (95% CI) |
| Province | Koshi | 2.47(1.38–4.42)** | 2.23(1.22–4.08)** |
| | Madhesh | 1.37(0.75–2.49) | 1.19(0.64–2.21) |
| | Gandaki | 1.76(0.86–3.57) | 1.60(0.78–3.31) |
| | Lumbini | 1.56(0.86–2.81) | 1.39(0.76–2.57) |
| | Karnali | 2.47(1.23–4.94)* | 2.28(1.11–4.70)* |
| | Sudurpaschim | 4.92(2.70–8.95)*** | 4.49(2.39–8.42)*** |
| | Bagmati | Reference | Reference |
| Area | Rural | 1.38(0.99–1.93) | 1.26(0.89–1.78) |
| | Urban | Reference | Reference |
| Age of Child | 7–23 months | | 1.56(1.10–2.20)* |
| | 0–6 months | | Reference |
| Sex of child | Male | | 1.21(0.91–1.61) |
| | Female | | Reference |
| ARI | Present | | 4.14(2.21–7.72)*** |
| | Absent | | Reference |
| Received prenatal care | No | | 1.87(1.01–3.45)* |
| | Yes | | Reference |
| Wealth group | Below richest | | 1.76(1.01–3.08)* |
| | Richest | | Reference |
| Sanitation | Limited/ Unimproved/open defecation | | 1.52(1.09–2.11)* |
| | Basic | | Reference |
| Hygiene | Limited/ No facility | | 0.87(0.61–1.24) |
| | Basic | | Reference |

Note: ***$p<0.001$ **$p<0.01$ *$p<0.05$; [†]Best fitting model; AOR: Adjusted Odds Ratio; CI: Confidence Interval; ICC: Null model 23%, Model 1: 17%, Model 2: 17%; -2 log-likelihood: Null model: 1662.33, Model 1: 1622.19, Model 2: 1578.63; Values based on SAS PROC GLIMMIX; Estimation method = quad

This study revealed that the risk of diarrhea was higher among children who had ARI symptoms compared with children who didn't have ARI symptoms. Another study has shown that the comorbidity of diarrhea and ARI can be both simultaneous or sequential [18]. Walker et al. discovered that diarrhea and ARI were present as simultaneous comorbidity in children aged under 5 years and that their correlation was stronger with the severity of the condition [38]. On the other hand, diarrhea might elevate the risk of ARI by creating a significant loss of micronutrients and dehydration thereby, weakening the immunity of children and predisposing a substantial risk of infection.

Our results also showed that the odds of diarrhea among under-two children were significantly higher among those whose mothers didn't receive prenatal care as compared to their counterparts which is supported by another study [21]. Mothers receive counseling on breastfeeding during their prenatal visits. Studies have shown that there is an increase in the rate and initiation of breastfeeding among women who received prenatal care [39, 40] which plays a significant role in lowering the risk of diarrheal morbidity [16]. Moreover, mothers who received prenatal care might increase their awareness of the common childhood illness and its management through counseling during their prenatal visits to health institutions.

Children from households with limited or unimproved sanitation or open defecation were more likely to suffer from diarrhea than those from households with basic sanitation, which was congruent with studies from Nepal [41] and Ethiopia [42, 43]. A study showed that the

lack of availability of a latrine was significantly associated with childhood diarrhea [19], as there is a high chance of water contamination when there is open defecation. Moreover, another study from Nepal reported that sharing a toilet is associated with the incidence of children diarrhea [26]. A possible explanation for our findings could be that when toilets are shared, there might be negligence of cleanliness responsibilities among shared households. Studies examining the issues of shared bathroom hygiene compared to private facilities have highlighted the necessity of duty rotas and responsibility for improved toilet access and use [44].

Furthermore, our study showed a higher risk of diarrhea morbidity among children from poor households compared to the richest household. The finding of our study corroborates the results of another study [45]. Wealthy households are more likely to use soap for hand washing, install water purifiers in their homes to prevent microbial contamination of water, and construct improved and hygienic toilets. On the contrary, lower-income households might not afford these facilities, which puts their children at higher risk of diarrhea morbidity. Various WASH interventions have shown diarrhea risk reductions in children [46].

## Association with external environmental variables

This study found that there is an uneven distribution of diarrhea across provinces. Children from Koshi Province, Karnali Province and Sudurpaschim Province have a higher risk of diarrheal morbidity compared with Bagmati Province. Karnali Province had the highest incidence of diarrhea per thousand children under five in 2020, followed by Sudurpaschim Province, whereas it was lowest in Bagmati Province [10]. Similar trends have been seen in previous fiscal years as well [11]. A multicountry study has shown that children who live in developing countries with greater per capita GDP were less likely to develop diarrhea [47]. Bagmati Province contributes the most to the overall GDP contribution among the seven provinces in Nepal [48]. The province's GDP reflects their residents' Per Capita Income. As discussed earlier, wealthy households could afford better WASH facilities, had better access to and use health services compared to poor households.

## Policy implications from this study

The findings of this study highlight disparities in the risk of diarrheal morbidity among under-two years children in Nepal, particularly among different age categories, socioeconomic and geographical groups. Interventions such as increasing awareness on the importance of complementary feeding hygiene, as well as promoting healthy feeding practices among mothers and caregivers can be implemented to reduce the risk of diarrhea among children aged 7–23 months. Furthermore, local governments in the Karnali and Sudurpaschim Provinces should focus on providing targeted interventions to children from poor families in collaboration with different stakeholders. These interventions should aim to increase access to health services, improve sanitation and hygiene condition through the provision of WASH facilities, and work towards improving the socio-economic status of the marginalized and vulnerable population in these areas.

## Strength and limitation

This study employed data from a recent nationally representative survey that used a standardized questionnaire and used a multi-level modeling method to account for the hierarchical nesting of complicated survey data. However, this study was unable to infer a causal association between diarrhea and associated factors because of its cross-sectional nature. However, given the established connection between diarrhea and poor sanitation facilities and feeding

behaviors, it does not affect our scientific merit much and it is still necessary to advocate the policymakers to mobilize resources to change the situation. Recall and reporting bias is likely because the study tools employed in this study analyzed prior habits and activities retrospectively.

## Conclusion and recommendations

The prevalence of diarrhea among children under 24 months was alarmingly high in Nepal. Interventional programs should focus on improving sanitation facilities, especially emphasizing poor households from Karnali and Sudurpaschim Provinces practicing open defecation to reduce the diarrheal burden in Nepal. In the future, we may conduct multicountry interventional studies to verify the causal roles of the factors associated with diarrhea.

## Acknowledgments

The authors would like to express our gratitude to Multiple Indicator Cluster Survey for allowing us to access and use the data set for the study.

## Author Contributions

**Conceptualization:** Bipin Thapa.

**Data curation:** Bipin Thapa.

**Formal analysis:** Bipin Thapa.

**Investigation:** Bipin Thapa.

**Methodology:** Bipin Thapa.

**Software:** Bipin Thapa.

**Visualization:** Bipin Thapa.

**Writing – original draft:** Bipin Thapa, Suman Sapkota.

**Writing – review & editing:** Bipin Thapa, Suman Sapkota, Yifei Hu.

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
