## [Decision Letter · Decision Letter 0]

19 Dec 2022

PONE-D-22-29049

Predictors of diarrhea among children under 24 months in Nepal: A multilevel analysis of Multiple Indicator Cluster Survey 2019

PLOS ONE

Dear Dr. Thapa,

Thank you for submitting your manuscript to PLOS ONE. After careful consideration, we feel that it has merit but does not fully meet PLOS ONE’s publication criteria as it currently stands. The reviewers have raised a number of points which we believe would improve the manuscript and may allow a revised version to be published in PLOS one.Therefore, we invite you to submit a revised version of the manuscript that addresses the points raised during the review process.

We look forward to receiving your revised manuscript.

Kind regards,

Ammal Mokhtar Metwally, Ph.D (MD)

Academic Editor

PLOS ONE

Journal Requirements:

Reviewers' comments:

Reviewer's Responses to Questions

**Comments to the Author**

1. Is the manuscript technically sound, and do the data support the conclusions?

Reviewer #1: Yes

Reviewer #2: Yes

2. Has the statistical analysis been performed appropriately and rigorously? 

Reviewer #1: Yes

Reviewer #2: Yes

3. Have the authors made all data underlying the findings in their manuscript fully available?

Reviewer #1: Yes

Reviewer #2: Yes

4. Is the manuscript presented in an intelligible fashion and written in standard English?

Reviewer #1: Yes

Reviewer #2: Yes

5. Review Comments to the Author

Reviewer #1: Dear authors,

I reviewed your paper entitled "Predictors of diarrhea among children under 24 months in Nepal: A multilevel analysis

of Multiple Indicator Cluster Survey 2019". Your analysis have several policy implications in the child health in Nepal. Here I drafted my comments in three section: Summary ( My takeaways, no need to address) , Major comments and Minor comments.

Summary

The authors analyzed Nepal's Multiple Indicator Cluster Survey (MICS) 2019 to analyze the factors associated with incidence of diarrhea among children below 24 months. They found that the prevalence of diarrhea among children aged 0-24 months was 11.9%. The predictors for the incidence of diarrhea were province of residence, age group, incidence of ARI, access to prenatal care, household's wealth status, open defecation, and access to improved sanitation facilities.

Major comments

Introduction:

The rationale for the study is not strong enough. Especially, what is the gap in previous literature about the factors associated with childhood diarrhea among children aged 0-24 months? Or in another words, is this topic previously studied or not? If yes, what is the gap you are trying to fulfill? I recommend you to provide literature review related to your study and provide gaps in the literature, and show the contribution made by your study. You can address this issue in your fourth paragraph of introduction section.

Conceptual framework

• The authors said that they categorized the factors into three categories as presented in Figure 1. Is the conceptual framework based on previous study or any standard framework? If yes, please provide the reference.

• In terms of vaccination status, I recommend to be specific about vaccines that prevent diarrhea. For example, Rota virus vaccine, Rota virus is the most common cause of diarrhea in low- and middle-income countries. I recommend to analyze your data using such vaccines, whether you see any change in p-value? If you observe any difference, you can include it in your model, otherwise, you can describe it in method section stating that we also analyzed our data using the vaccination status of vaccines that prevent diarrhea.

• Is there any reason to include wasting only? Why you did not include stunting and underweight? Please describe about it while describing your study variables.

• Please provide reference while categorizing caste ethnic group into two groups: relatively advantaged and relatively disadvantaged.

Statistical Methods

• You said that those variables that have p-value less than 0.2 were included in multivariate analysis. Could you provide any reference that recommend this concept? or is this arbitrary cut-off?

• I think your final model is model 2, but it is not clear in data analysis section, please make it clear in words.

Discussion

• In line number 238: You provided association of GDP and diarrhea to explain the possible linkage of Province and incidence of diarrhea. Please support your argument with reference that shows the association between GDP and incidence of diarrhea.

Conclusion and Recommendations

• What are the policy implications of your study? You only talked about it in conclusion section, I think you need a separate subsection for policy implications of your findings. What are the 2-3 key points that policymakers can take from your study?

Minor Comments

• In table 4, please use reference instead of "1" as a reference category.

• Title of table 2: Characteristics of children under 24 months in seven provinces of Nepal. It does make sense to use "seven provinces of Nepal". You can simple use sample characteristics or sample characteristics of children under 24 months.

Reviewer #2: manuscript is well written by utilizing the data from the open access data of MICS conducted in Nepal. I suggest authors to more minutely discuss the "participation and sample size " section as it looks inadequate and incomplete. Please specify "what was the basis for identifying numbers of urban and rural strata? you have mentioned "Within each stratum, primary

81 sampling units (PSUs) were selected systematically with probability proportional to size." how did you decide to take 81 PSUs? how did you calculated the sample size? please elaborate.

6. PLOS authors have the option to publish the peer review history of their article (what does this mean?). If published, this will include your full peer review and any attached files.

Reviewer #1: No

Reviewer #2: **Yes: **Dr Kishor Adhikari

---

## [Author Response · Author response to Decision Letter 0]

9 Feb 2023

Reviewer #1: Major comments

Reviewer: The rationale for the study is not strong enough. Especially, what is the gap in previous literature about the factors associated with childhood diarrhea among children aged 0-24 months? Or in another words, is this topic previously studied or not? If yes, what is the gap you are trying to fulfill? I recommend you to provide literature review related to your study and provide gaps in the literature, and show the contribution made by your study. You can address this issue in your fourth paragraph of introduction section.

Authors: Thank you very much for the suggestion. We have added the current literature and gaps in the introduction section.

Reviewer: The authors said that they categorized the factors into three categories as presented in Figure 1. Is the conceptual framework based on previous study or any standard framework? If yes, please provide the reference.

Authors: Thank you very much for the question. The conceptual framework is not based on solely the previous study or framework. It is based on an extensive literature review. After the extensive literature review, we categorized the significant variable available in the dataset into three categories as presented in the Figure 1. We have added few lines in the conceptual framework subsection to make it clearer. 

Reviewer: In terms of vaccination status, I recommend to be specific about vaccines that prevent diarrhea. For example, Rota virus vaccine, Rota virus is the most common cause of diarrhea in low- and middle-income countries. I recommend to analyze your data using such vaccines, whether you see any change in p-value? If you observe any difference, you can include it in your model, otherwise, you can describe it in the method section stating that we also analyzed our data using the vaccination status of vaccines that prevent diarrhea.

Authors: Thank you very much for the suggestion. The data was taken from the Multiple Indicator Cluster Survey 2019, where there is a variable named “Rotavirus vaccine ever given”. Under this variable, it is mentioned that 27 under 24-months children were given Rotavirus vaccination, 358 under 24-month children were not given Rotavirus vaccination, 142 reported “don’t know”, 1 didn’t respond and 2039 observations were missing. Therefore, in order to avoid possible unreliable parameter estimation and larger confidence interval due to inadequate sample size for the Rotavirus vaccine ever given variable, the authors did not include this variable in the study. 

Reviewer: Is there any reason to include wasting only? Why you did not include stunting and underweight? Please describe about it while describing your study variables.

Authors: Thank you very much for the question. After the extensive literature review, we found that wasting among these three indicators was the most appropriate factor responsible for childhood diarrhea. Wasting was the leading risk factor responsible for 80.4% (95% UI 68.2–85.0) of diarrhea deaths in children younger than 5 years according to recent studies. Therefore, we included only wasting as the indicator of malnutrition to make the model parsimonious as mentioned in the analysis subsection. 

Reference: Troeger, C.; Blacker, B.F.; Khalil, I.A.; Rao, P.C.; Cao, S.; Zimsen, S.R.; Albertson, S.B.; Stanaway, J.D.; Deshpande, A.; Abebe, Z.; et al. Estimates of the Global, Regional, and National Morbidity, Mortality, and Aetiologies of Diarrhoea in 195 Countries: A Systematic Analysis for the Global Burden of Disease Study 2016. Lancet Infect. Dis. 2018, 18, 1211–1228, doi:10.1016/S1473-3099(18)30362-1/ATTACHMENT/35A61B31-B1D1-4B4D-9020-

Reviewer: Please provide reference while categorizing caste ethnic group into two groups: relatively advantaged and relatively disadvantaged.

Authors: Thank you very much for the suggestion. It was categorized based on the report: Trends and determinants of neonatal mortality in Nepal: Further analysis of the Nepal Demographic and Health Surveys, 2001-2011. The categorization criteria have been reported on page 5 of before mentioned document. We have added the reference in the manuscript. 

Reference: Paudel D, Thapa A, Shedain P, Paudel B. Trends and determinants of neonatal mortality in Nepal: Further analysis of the Nepal Demographic and Health Surveys, 2001-2011. 2013. 

Reviewer: You said that those variables that have p-value less than 0.2 were included in multivariate analysis. Could you provide any reference that recommend this concept? or is this arbitrary cut-off?

The level of significance was set at 0.05 for both bivariate and multivariable analysis. However, we selected the variables which had p-value less than 0.2 in bivariate analysis to be included in the multivariable analysis. It was done so that the important variables which were not significant at traditional level of significance (0.05) could also be included in the model. The fit statistics of models were found to be more appropriate when including variables with p-value less than 0.2 in the model in comparison to model with variables having p-value less than 0.05. Moreover, a study conducted by Bursac et al., has discussed that in order to improve the chances of retaining meaningful confounders, the confounding level can be set above 5%.

Reference: Bursac, Z., Gauss, C.H., Williams, D.K., Hosmer, D.W., 2008. Purposeful selection of variables in logistic regression. Source Code Biol. Med. 2008 31 3, 1–8. https://doi.org/10.1186/1751-0473-3-17

Reviewer: I think your final model is model 2, but it is not clear in data analysis section, please make it clear in words.

Authors: Thank you very much for your comment. Yes, model 2 is the final model. We have split the sentence and added some words to make it clear in the data analysis section.

Reviewer: In line number 238: You provided association of GDP and diarrhea to explain the possible linkage of Province and incidence of diarrhea. Please support your argument with reference that shows the association between GDP and incidence of diarrhea.

Authors: Thank you very much for your suggestion. We have added the statement that shows the association between GDP and the incidence of diarrhea in the discussion section to support our argument.

Reviewer: What are the policy implications of your study? You only talked about it in conclusion section, I think you need a separate subsection for policy implications of your findings. What are the 2-3 key points that policymakers can take from your study?

Authors: Thank you very much for the suggestion. As per your suggestion, we have added a different subsection for the policy implication of our study above the limitation section. 

Reviewer #1: Minor comments

Reviewer: In table 4, please use reference instead of "1" as a reference category.

Authors: As per your suggestion, we have changed 1 to reference in table 4.

Reviewer: Title of table 2: Characteristics of children under 24 months in seven provinces of Nepal. It does make sense to use "seven provinces of Nepal". You can simple use sample characteristics or sample characteristics of children under 24 months.

Authors: We have changed the title of tables as per your suggestion.

Reviewer #2: Manuscript is well written by utilizing the data from the open access data of MICS conducted in Nepal. I suggest authors to more minutely discuss the "participation and sample size " section as it looks inadequate and incomplete. Please specify "what was the basis for identifying numbers of urban and rural strata? you have mentioned "Within each stratum, primary 81 sampling units (PSUs) were selected systematically with probability proportional to size." how did you decide to take 81 PSUs? how did you calculated the sample size? please elaborate.

Author: Thank you very much for the question. As per your suggestion, we have added few lines under “Participants and Sample Size” to make it more clearer, Since, our study adopted the dataset from Multiple Indicator Cluster Survey 2019, which was conducted by the Central Bureau of Statistics, the sampling and sample size calculation was performed by Central Bureau of Statistics, National Planning Commission, Government of Nepal. A multi-stage, stratified cluster sampling approach was used for the selection of the survey sample. The urban and rural areas within each province were made main sampling strata. Rural municipalities represent the rural area whereas the municipalities, sub-metropolitan, and metropolitan cities represent the urban area. Within each stratum, primary sampling units (PSUs) were selected systematically with probability proportional to size. The PSUs selected at the first stage were the enumeration areas (EAs) defined for the census enumeration. A listing of households was conducted in each sample EA, and a sample of households was selected at the second stage.

Since the overall sample size for the Nepal MICS partly depends on the geographic domains of analysis, the distribution of EAs and households in Nepal from the 2011 Census sampling frame was first examined by province, urban and rural strata. As In the survey total of 14,805 women of age 15-49 years were interviewed, which included 6658 mothers/caretakers of children under 5 years. After receiving the dataset, observations of children above 23 months were excluded and 2566 observations out of 6658 mothers/caretakers were retained. Furthermore, 218 observations for different variables with missing data were removed and 2348 observations nested into 512 PSUs were included in the final analysis. 

Reference: Central Bureau of Statistics (CBS), 2020. Nepal Multiple Indicator Cluster Survey 2019, Survey Findings Report. Kathmandu, Nepal: Central Bureau of Statistics and UNICEF Nepal.

---

## [Decision Letter · Decision Letter 1]

20 Apr 2023

PONE-D-22-29049R1Predictors of diarrhea among children under 24 months in Nepal: A multilevel analysis of Multiple Indicator Cluster Survey 2019PLOS ONE

Dear Dr. Thapa,

Thank you for submitting your manuscript to PLOS ONE. After careful consideration, we feel that it has merit but does not fully meet PLOS ONE’s publication criteria as it currently stands. Therefore, we invite you to submit a revised version of the manuscript that addresses the points raised during the review process.

We look forward to receiving your revised manuscript.

Kind regards,

Ammal Mokhtar Metwally, Ph.D (MD)

Academic Editor

PLOS ONE

Journal Requirements:

Reviewers' comments:

Reviewer's Responses to Questions

**Comments to the Author**

1. If the authors have adequately addressed your comments raised in a previous round of review and you feel that this manuscript is now acceptable for publication, you may indicate that here to bypass the “Comments to the Author” section, enter your conflict of interest statement in the “Confidential to Editor” section, and submit your "Accept" recommendation.

Reviewer #1: All comments have been addressed

Reviewer #2: All comments have been addressed

2. Is the manuscript technically sound, and do the data support the conclusions?

Reviewer #1: Yes

Reviewer #2: Yes

3. Has the statistical analysis been performed appropriately and rigorously? 

Reviewer #1: Yes

Reviewer #2: Yes

4. Have the authors made all data underlying the findings in their manuscript fully available?

Reviewer #1: Yes

Reviewer #2: Yes

5. Is the manuscript presented in an intelligible fashion and written in standard English?

Reviewer #1: Yes

Reviewer #2: Yes

6. Review Comments to the Author

Reviewer #1: Dear Authors,

Thank you for addressing my comments. Here are my additional comments. I hope this paper will be better after addressing these comments.

Comments:

1. The sentence mentioned below is very strong as if there are no studies that explored diarrheal risk in under two children. Please revise your sentence because there are some studies that have explored heterogeneity in diarrheal risk across age groups while studying diarrhea in under five children.

"Although previous studies reported that the diarrheal risk is higher in under two years children, no studies have been conducted in this specific population"

It could be: To the extent of our knowledge, we did not find national-level studies that explicitly explored diarrheal risk in under two years children in Nepal.

2. As suggested in first review, I think you have not provided the reference for determining the p-value of <0.2?

"First, bivariate analysis was conducted where the Rao-Scott chi131 square test was applied to determine the p-value between diarrhea and independent variables, and variables significant at p<0.2 in bivariate analysis were considered candidate variables for the multilevel analysis"

3. Please correct "Province one" as "Koshi Province" in table and other places. as this province is recently named.

4. Is this policy implications based on the findings of your study? "It is important to note that these interventions need to be culturally appropriate, and involve community participation to ensure their sustainability." If not, provide reference and try to link with your study's findings.

5. Conceptual framework title: I think it should be like this: Conceptual framework for diarrheal risk among under two children

Thank you.

Best,

Reviewer

Reviewer #2: All the concerns are addressed and modified as per the suggestion. I am happy with the this version of manuscript.

7. PLOS authors have the option to publish the peer review history of their article (what does this mean?). If published, this will include your full peer review and any attached files.

Reviewer #1: No

Reviewer #2: **Yes: **Dr. Kishor Adhikari

---

## [Author Response · Author response to Decision Letter 1]

30 Apr 2023

Dear reviewer, Thank you for your valuable comments. We have addressed all the comments and incorporated your suggestion in the revised version of the manuscript. 

Comments: 

1. The sentence mentioned below is very strong as if there are no studies that explored diarrheal risk in under two children. Please revise your sentence because there are some studies that have explored heterogeneity in diarrheal risk across age groups while studying diarrhea in under five children.

"Although previous studies reported that the diarrheal risk is higher in under two years children, no studies have been conducted in this specific population"

It could be: To the extent of our knowledge, we did not find national-level studies that explicitly explored diarrheal risk in under two years children in Nepal.

Authors: Thank you very much for your suggestion. We have changed the sentence according to your suggestion in line number 74-76.

2. As suggested in first review, I think you have not provided the reference for determining the p-value of <0.2?

"First, bivariate analysis was conducted where the Rao-Scott chi131 square test was applied to determine the p-value between diarrhea and independent variables, and variables significant at p<0.2 in bivariate analysis were considered candidate variables for the multilevel analysis"

Authors: Thank you very much for your suggestion. We have added the reference as per your suggestion. The words “variables significant at p<0.2 in bivariate analysis were considered candidate variables for the multilevel analysis” have been changed to “variables having p-value less than 0.2 in bivariate analysis were considered candidate variables for the multilevel analysis” in lines number 132 and 133 in the revised version of the manuscript for more clarity. 

3. Please correct "Province one" as "Koshi Province" in table and other places. as this province is recently named.

Authors: Thank you so much for your suggestion. We have changed "Province One" to "Koshi Province" throughout the manuscript.

4. Is this policy implications based on the findings of your study? "It is important to note that these interventions need to be culturally appropriate, and involve community participation to ensure their sustainability." If not, provide reference and try to link with your study's findings.

Authors: Thank you so much for your valuable suggestion. We have deleted the lines as per your suggestion.

5. Conceptual framework title: I think it should be like this: Conceptual framework for diarrheal risk among under two children

Authors: Thank you so much for the suggestion. We have changed the conceptual framework’s title accordingly.

---

## [Decision Letter · Decision Letter 2]

19 Jun 2023

Predictors of diarrhea among children under 24 months in Nepal: A multilevel analysis of Multiple Indicator Cluster Survey 2019

PONE-D-22-29049R2

Dear Dr. Thapa,

We’re pleased to inform you that your manuscript has been judged scientifically suitable for publication and will be formally accepted for publication once it meets all outstanding technical requirements.

Kind regards,

Ammal Mokhtar Metwally, Ph.D (MD)

Academic Editor

PLOS ONE

Additional Editor Comments (optional):

Reviewers' comments:

Reviewer's Responses to Questions

**Comments to the Author**

1. If the authors have adequately addressed your comments raised in a previous round of review and you feel that this manuscript is now acceptable for publication, you may indicate that here to bypass the “Comments to the Author” section, enter your conflict of interest statement in the “Confidential to Editor” section, and submit your "Accept" recommendation.

Reviewer #1: All comments have been addressed

Reviewer #2: All comments have been addressed

2. Is the manuscript technically sound, and do the data support the conclusions?

Reviewer #1: Yes

Reviewer #2: Yes

3. Has the statistical analysis been performed appropriately and rigorously? 

Reviewer #1: Yes

Reviewer #2: Yes

4. Have the authors made all data underlying the findings in their manuscript fully available?

Reviewer #1: Yes

Reviewer #2: Yes

5. Is the manuscript presented in an intelligible fashion and written in standard English?

Reviewer #1: Yes

Reviewer #2: Yes

6. Review Comments to the Author

Reviewer #1: Dear Authors,

Thank you for addressing all of my comments. Hope my comments were helpful to improve the manuscript.

Best,

Reviewer

Reviewer #2: (No Response)

7. PLOS authors have the option to publish the peer review history of their article (what does this mean?). If published, this will include your full peer review and any attached files.

Reviewer #1: No

Reviewer #2: **Yes: **Dr. Kishor Adhikari

---

## [Editor Report · Acceptance letter]

26 Jun 2023

PONE-D-22-29049R2 

Predictors of diarrhea among children under 24 months in Nepal: A multilevel analysis of Multiple Indicator Cluster Survey 2019 

Dear Dr. Thapa:

I'm pleased to inform you that your manuscript has been deemed suitable for publication in PLOS ONE. Congratulations! Your manuscript is now with our production department. 

Kind regards, 

on behalf of

Professor Ammal Mokhtar Metwally 

Academic Editor

PLOS ONE